# Age Increases the Risk of Mortality by Four-Fold in Patients with Emergent Paralytic Ileus: Hospital Length of Stay, Sex, Frailty, and Time to Operation as Other Risk Factors

**DOI:** 10.3390/ijerph19169905

**Published:** 2022-08-11

**Authors:** Guy Elgar, Parsa Smiley, Abbas Smiley, Cailan Feingold, Rifat Latifi

**Affiliations:** 1Westchester Medical Center, School of Medicine, New York Medical College, Valhalla, NY 10595, USA; 2School of Engineering, University of Massachusetts at Amherst, Amherst, MA 01003, USA; 3Minister of Health, 10000 Pristina, Kosovo; 4School of Medicine, University of Arizona, Tucson, AZ 85721, USA

**Keywords:** hospital length of stay, time to operation, paralytic ileus, postoperative ileus, in-hospital mortality

## Abstract

**Background**: In the United States, ileus accounts for USD 750 million of healthcare expenditures annually and significantly contributes to morbidity and mortality. Despite its significance, the complete picture of mortality risk factors for these patients have yet to be fully elucidated; therefore, the aim of this study is to identify mortality risk factors in patients emergently admitted with paralytic ileus. **Methods**: Adult and elderly patients emergently admitted with paralytic ileus between 2005–2014 were investigated using the National Inpatient Sample Database. Clinical outcomes, therapeutic management, demographics and comorbidities were collected. Associations between mortality and all other variables were established via univariable and multivariable logistic regression models. **Results**: A total of 81,674 patients were included, of which 45.2% were adults, 54.8% elderly patients, 45.8% male and 54.2% female. The average adult and elderly ages were 48.3 and 78.8 years, respectively. Elderly patients displayed a significantly (*p* < 0.01) higher mortality rate (3.0%) than adults (0.7%). The final multivariable logistic regression model showed that for every one-day delay in operation, the odds of mortality for adult and elderly patients increased by 4.1% (*p* = 0.002) and 3.2% (*p* = 0.014), respectively. Every additional year of age corresponded to 3.8% and 2.6% increases in mortality for operatively managed adult (*p* = 0.026) and elderly (*p* = 0.015) patients. Similarly, non-operatively treated adult and elderly patients displayed associations between mortality and advanced age (*p* = 0.001). The modified frailty index exhibited associations with mortality in operatively treated adults, conservatively managed adults and conservatively managed elderly patients (*p* = 0.001). Every additional day of hospitalization increased the odds of mortality in non-operative adult and elderly patients by 7.6% and 5.8%, respectively. Female sex correlated to lower mortality rates in non-operatively managed adult patients (odds ratio = 0.71, *p* = 0.028). Undergoing invasive diagnostic procedures in non-operatively managed elderly patients related to reduced mortality (odds ratio = 0.78, *p* = 0.026). **Conclusions**: Patients emergently admitted for paralytic ileus with increased hospital length of stay, longer time to operation, advanced age or higher modified frailty index displayed higher mortality rates. Female sex and invasive diagnostic procedures were negatively correlated with death in nonoperatively managed patients with paralytic ileus.

## 1. Introduction

Ileus contributes to a high number of admissions, morbidity and mortality across the world. From a surgical perspective, postoperative ileus (POI) is one of the most common complications following abdominal and non-abdominal operations [1,2]. The current literature approximates the incidence of ileus after abdominal surgery to be between 10–27% [2,3,4]. Previous studies have indicated POI is the single greatest factor affecting hospital length of stay (HLOS) after bowel resection [5]. In addition, ileus significantly contributes to healthcare costs in the United States where the resulting economic stress is estimated to be USD 750 million annually [2]. Similarly, a large-scale analysis of economic consequences of POI revealed that the total cost of care for a single patient is significantly higher for patients experiencing POI relative to those who do not develop POI (USD 18,000 vs. USD 11,700, respectively) [6]. These findings illustrate that ileus significantly contributes to healthcare-associated economic burden and constitutes one of the most prevalent postoperative complications. While ileus generally has a low mortality rate of between 2–10%, the volume of patients experiencing this ailment indicates a significant sum of lives are at risk [7,8].

The pathogenesis of ileus is very complex, including a number of factors that have yet to be fully elucidated. Most analyses describe various influences including, but not limited to, gastrointestinal stretch, fluid balance, opioids, neuro-hormonal factors and inflammation [9]. Despite a large number of clinical studies, treatment protocols and prophylactic strategies have yet to be established [10]. While previous analyses have contributed useful information, further research involving scoring tools, clinical treatment patterns and patient characteristics are necessary to improve prognostication in patients with paralytic ileus. It is essential to identify the risk factors of mortality in paralytic ileus in order to allow physicians to identify higher-risk patients and to allow the field to mitigate the cost of paralytic ileus, both in dollars and in lives. We suspect that the patient characteristics of age and frailty are significant risk factors for mortality, as both have been associated with complications and mortality in both surgical and non-surgical patient populations. Additionally, we hypothesize that longer HLOS is associated with an increased risk of mortality. Finding risk factors specific to paralytic ileus will allow healthcare providers to better manage these patients and reduce the burden of this condition.

The primary aim of this study is to identify potential predictors of mortality in emergently admitted patients with the primary diagnosis of paralytic ileus through a 10-year retrospective analysis of 81,674 patients.

## 2. Methods

The retrospective analysis discussed above was based upon data compiled from the National Inpatient Sample (NIS) repository for patients who were emergently admitted with a primary diagnosis of paralytic ileus (International Classification of Diseases (ICD)-0 code 560.1) from 2005 to 2014.

The Agency for Healthcare Research and Quality funded the Healthcare Cost and Utilization Project to create nationally obtained population-based data in standardized formats. This sponsorship led to the creation of the NIS, a government-supported database. Information included within the NIS allows researchers to complete large-scale analyses that incorporate a variety of biological, societal and psychological factors. These investigations are highly generalizable and therefore provide excellent discernment of the relationships between clinical outcomes, patient characteristics, therapeutic protocols and epidemiological trends.


Patient Features


The results analyzed in this study were stratified according to the characteristics of interest, including age differences, operation status (operation vs. no operation), sex categories, types of operation or diagnostic procedure utilized, and clinical outcomes (survived or deceased). Patient features were further evaluated based upon race (White, Black, Hispanic, Asian/Pacific Islander, Native American and other), income quartile, insurance status (private insurance, Medicare, Medicaid, self-pay, no charge and other), hospital location (rural, urban non-teaching and urban teaching), comorbidities (AIDS, alcohol abuse, deficiency anemias, rheumatoid arthritis, chronic blood loss, congestive heart failure, chronic pulmonary disease, coagulopathy, depression, uncomplicated diabetes, chronic complicated diabetes, drug abuse, hypertension, hypothyroidism, liver disease, lymphoma, fluid/electrolyte disorders, metastatic cancer, other neurological disorders, obesity, paralysis, peripheral vascular disorders, psychoses, pulmonary circulation disorders, renal failure, presence of solid tumor, peptic ulcer disease, valvular disease and weight loss), invasive diagnostic procedure, surgical procedure, invasive or surgical procedure, hospital length of stay, total charges in dollars, time to surgical procedure, time to invasive diagnostic procedure and modified frailty index. Surgical procedures included the following: operations on the esophagus, stomach, intestines, appendix, rectum, rectosigmoid, and perirectal tissue, anus, liver, gallbladder and biliary tract, pancreas, hernia and other abdominal regions (Table 1). Invasive diagnostic procedures included procedures on the esophagus, stomach, intestine, rectum, rectosigmoid, and perirectal tissue, anus, liver, gallbladder and biliary tract, pancreas and other abdominal regions (Table 1). International Classification of Diseases (ICD)-9 relating to specific operations or invasive diagnostic procedures are shown in Table 1. It is important to note that the NIS does not contain all variables required to calculate the 5-item modified frailty index, nor does it specify the duration of the comorbidity prior to admission. Therefore, to include frailty in the study, we estimated the 5-item modified frailty index with the variables available in the dataset. Diabetes was estimated to be present if either the comorbidity of uncomplicated diabetes or diabetes with chronic complications was indicated. A history of congestive heart failure was assumed to be present if congestive heart failure comorbidity was indicated. The use of medication to control hypertension could not be determined from the data set; it was estimated by the indication of hypertension. A history of chronic obstructive pulmonary disease (COPD) was estimated using the comorbidity of chronic pulmonary disease. Functional health status was not available in the dataset and had to be estimated based on the available variables. If the patient had a comorbidity of tumor, renal failure, metastatic cancer, paralysis, lymphoma, coagulopathy or weight loss, it was assumed they were partially or totally dependent and therefore functionally dependent. Both instruments assigned one point to each item if the comorbidity item was present. The items were summed to create an estimated modified frailty index. The scale ranged from 0 to 5, with 0 being not frail and 5 being very frail. The subsequent result was recorded as an average (SD) within the modified frailty index row in Table 2, Table 3, Table 4, Table 5 and Table 6. Table 7 illustrates the NIS variables utilized in our modified 5-item frailty index as they correspond to the Subramaniam 5-item modified frailty index factors.

**Table 1 ijerph-19-09905-t001:** Characteristics of emergency-admitted patients with the primary diagnosis of paralytic ileus. Data are stratified according to surgical procedure (NIS 2005–2014).

Digestive Surgical Procedure (ICD 9)	Adults, N (%)				Elderly, N (%)			
	Procedures	Survived	Deceased	*p*	Procedures	Survived	Deceased	*p*
Operations on Esophagus (42.01–42.19, 42.31–42.99)	47 (2.0%)	46 (97.9%)	1 (2.1%)	0.270	104 (3.6%)	98 (94.2%)	6 (5.8%)	0.100
Operations on Stomach (43.0–44.03, 44.21–44.99)	235 (10.0%)	222 (94.5%)	13 (5.5%)	**<0.001**	392 (13.4%)	359 (91.6%)	33 (8.4%)	**<0.001**
Operations on Intestine (45.00–45.03, 45.30–46.99)	723 (30.6%)	698 (96.5%)	25 (3.5%)	**<0.001**	1312 (44.9%)	1215 (92.6%)	97 (7.4%)	**<0.001**
Operations on Appendix (47.01–47.99)	124 (5.3%)	123 (99.2%)	1 (0.8%)	0.570	41 (1.4%)	39 (95.1%)	2 (4.9%)	0.350
Operations on Rectum, Rectosigmoid and Perirectal Tissue (48.0–48.1, 48.31–48.99)	60 (2.5%)	60 (100%)	0 (0%)	0.999	97 (3.3%)	92 (94.8%)	5 (5.2%)	0.220
Operations on Anus (49.01–49.12, 49.31–49.99)	7 (0.3%)	7 (100%)	0 (0%)	0.999	12 (0.4%)	11 (91.7%)	1 (8.3%)	0.310
Operations on Liver (50.0, 50.21–50.99)	7 (0.3%)	7 (100%)	0 (0%)	0.999	3 (0.1%)	3 (100%)	0 (0%)	0.999
Operations on Gallbladder and Biliary Tract (51.01–51.04, 51.21–51.99)	118 (5.0%)	110 (93.2%)	8 (6.8%)	**<0.001**	163 (5.6%)	152 (93.3%)	11 (6.7%)	**0.006**
Operations on Pancreas (52.01–52.09, 52.21–52.99)	5 (0.2%)	4 (80.0%)	1 (20.0%)	**0.033**	2 (0.07%)	2 (100%)	0 (0%)	0.999
Operations on Hernia (53.00–53.9)	62 (2.6%)	61 (98.4%)	1 (1.6%)	0.340	72 (2.5%)	66 (91.7%)	6 (8.3%)	**0.009**
Other Operations on Abdominal Region (54.0–54.19, 54.3–54.99)	971 (41.2%)	929 (95.7%)	42 (4.3%)	**<0.001**	726 (24.8%)	641 (88.3%)	85 (11.7%)	**<0.001**
**Digestive Invasive Diagnostic Procedure (ICD 9)**	**Procedures**	**Survived**	**Deceased**	** *p* **	**Procedures**	**Survived**	**Deceased**	** *p* **
Invasive Diagnostic Procedure on Esophagus (42.21–42.29)	9 (0.2%)	9 (100%)	0 (0%)	0.999	9 (0.2%)	8 (88.9%)	1 (11.1%)	0.240
Invasive Diagnostic Procedure on Stomach (44.11–44.19)	12 (0.3%)	12 (100%)	0 (0%)	0.999	18 (0.4%)	17 (94.4%)	1 (5.6%)	0.430
Invasive Diagnostic Procedure on Intestine (45.11–45.29)	3397 (88.7%)	3365 (99.1%)	32 (0.9%)	**0.048**	4372 (91.8%)	4226 (96.7%)	146 (3.3%)	0.210
Invasive Diagnostic Procedure on Rectum, Rectosigmoid and Perirectal Tissue (48.21–48.29)	136 (3.6%)	134 (98.5%)	2 (1.5%)	0.240	158 (3.3%)	154 (97.5%)	4 (2.5%)	0.999
Invasive Diagnostic Procedure on Anus (49.21–49.29)	6 (0.2%)	6 (100%)	0 0(%)	0.999	5 (0.1%)	5 (100%)	0 (0%)	0.999
Invasive Diagnostic Procedure on Liver (50.11–50.19)	43 (1.1%)	42 (97.7%)	1 (2.3%)	0.250	36 (0.8%)	34 (94.4%)	2 (5.6%)	0.300
Invasive Diagnostic Procedure on Gallbladder and Biliary Tract (51.10–51.19)	21 (0.5%)	20 (95.2%)	1 (4.8%)	0.130	23 (0.5%)	22 (95.7%)	1 (4.3%)	0.510
Invasive Diagnostic Procedure on Pancreas (52.11–52.19)	1 (0.03%)	1 (100%)	0 (0%)	0.999	5 (0.1%)	4 (80.0%)	1 (20.0%)	0.140
Invasive Diagnostic Procedure on Other Operations on Abdominal Region (54.21–54.29)	204 (5.3%)	201 (98.5%)	3 (1.5%)	0.160	135 (2.8%)	126 (93.3%)	9 (6.7%)	**0.014**

Bold indicates result is statistically significant.


Statistical Analysis:


Following stratification, the data were applied in more complex investigations shown as Tables. The findings are shown alongside the average, confidence interval set at 95% (CI), *p*-values (0.05 or less considered significant) and standard deviation (SD). Univariable and multivariable logistic regression analyses were performed to determine the associations between mortality and parameters of interest. A backward logistic regression analysis was completed utilizing stepwise backward elimination to evaluate mortality and risk factors while controlling for confounding variables. We assessed the correlations among all independent variables. In none of them was the correlation coefficient greater than 0.3. These analyses were completed using SPSS version 24 (SPSS Inc., Chicago, IL, USA). R software (Foundation for Statistical Computing, Vienna, Austria) was used to assess the possible non-linear association of continuous variables, such as HLOS and age, with mortality through a generalized additive model (GAM package).

## 3. Results

### 3.1. Surgical and Invasive Diagnostic Procedures

Table 1 represents a stratified analysis regarding survival following surgical and invasive diagnostic procedures in adult and elderly patients with the primary diagnosis of paralytic ileus. In the elderly population, intestinal operations (44.9%) made up the majority of surgeries, followed by other operations on abdominal regions (24.8%) and stomach operations (13.4%). Adults underwent other operations on abdominal regions at the highest rate (41.2%), followed by intestinal operations (30.6%) and stomach operations (10.0%). The operations associated with the highest mortality rate in elderly patients were other operations on abdominal (11.7%); stomach (8.4%); hernia (8.3%); anal (8.3%); intestinal (7.4%); gallbladder and biliary tract (6.7%); esophageal (5.8%), rectal, rectosigmoid and perirectal (5.2%) and appendicular (4.9%) regions. No elderly patient deaths were recorded following pancreatic and liver operations. Adults experienced the highest mortality rates when undergoing the following operations: pancreatic (20.0%), gallbladder and biliary tract (6.8%), stomach (5.5%), other operations on abdominal regions (4.3%), intestinal (3.5%), esophageal (2.1%), hernia (1.6%) and appendicular (0.8%). No adult patient deaths occurred following rectal, rectosigmoid and perirectal; anal; and liver operations.

Intestinal invasive diagnostic procedures accounted for 91.8% and 88.7% of diagnostic procedures in elderly and adult patients, respectively (Table 1). The invasive diagnostic procedures associated with the highest mortality rates in elderly patients were pancreatic (20.0%), esophageal (11.1%), other operations on abdominal region (6.7%), stomach (5.6%), liver (5.6%), gallbladder and biliary tract (4.3%), intestinal (3.3%) and rectal; rectosigmoid; and perirectal (2.5%) procedures. No elderly patients died following invasive diagnostic procedures on the anus. Adults experienced the highest mortality rates when undergoing the following invasive diagnostic procedures: gallbladder and biliary tract (4.8%); liver (2.3%); rectal, rectosigmoid and perirectal (1.5%); other operations on abdominal region (1.5%) and intestinal (0.9%). No adult patient deaths occurred following esophageal, stomach, anal or pancreatic procedures.

### 3.2. Gender Differences

During 2005–2014, a total of 81,674 patients were admitted emergently with the diagnosis of ileus, including 36,916 adult (aged 18–64 years) and 44,758 elderly (aged 65+ years) patients. A total of 17,702 were adult males (48.0%), 19,214 were adult females (52.0%), 19,740 were elderly males (44.1%) and 25,018 were elderly females (55.9%). Adults had a similar mean age of 48 years (Table 2), while elderly males had a mean age of 77 years and elderly women had a mean age of 79 years. Among the adults, the mean (SD) age of the 250 (0.7%) patients who died during the study period was 53.98 (9.03) years, out of which 146 were males (58.4%) and 104 were females (41.6%). Regardless of gender, most adult patients were White, funded by private insurance and admitted to urban non-teaching hospitals (Table 2). Among the elderly, the mean (SD) age of the 1356 (3.0%) patients who died during the study period was 81.54 (8.01) years; 615 were males (45.4%) and 741 were females (54.6%). Regardless of gender, most elderly patients were White, funded by Medicare and were admitted to urban non-teaching hospitals (Table 2). Major comorbidities were hypertension, deficiency anemias, depression, uncomplicated diabetes and fluid/electrolyte disorders. Adult males showed higher rates of alcohol abuse, coagulopathy, liver disease, uncomplicated diabetes and drug abuse, while adult females showed higher rates of deficiency anemias, chronic pulmonary disease, hypothyroidism, fluid/electrolyte disorders and obesity. However, among the elderly females, chronic pulmonary disease rates were lower compared to elderly males. Adult females had higher rates of invasive diagnostic procedures, but lower rates of surgical procedures compared to adult males. Elderly females had lower rates of invasive diagnostic, surgical and invasive procedures compared to elderly men (Table 2).

**Table 2 ijerph-19-09905-t002:** Characteristics of emergency-admitted patients with the primary diagnosis of paralytic ileus (NIS 2005–2014). Data are stratified according to sex categories.

		Adult, N (%)			Elderly, N (%)		
		Male	Female	*p*	Male	Female	*p*
**All Cases**		17,702 (48.0%)	19,214 (52.0%)		19,740 (44.1%)	25,018 (55.9%)	
**Race**	**White**	10,485 (70.0%)	11,357 (70.7%)	**0.002**	13,082 (79.1%)	16,717 (80.3%)	**0.002**
	**Black**	2493 (16.6%)	2708 (16.9%)		1758 (10.6%)	2053 (9.9%)	
	**Hispanic**	1302 (8.7%)	1348 (8.4%)		987 (6.0%)	1267 (6.1%)	
	**Asian/Pacific Islander**	245 (1.6%)	243 (1.5%)		309 (1.9%)	314 (1.5%)	
	**Native American**	102 (0.7%)	120 (0.7%)		91 (0.6%)	127 (0.6%)	
	**Other**	357 (2.4%)	278 (1.7%)		309 (1.9%)	330 (1.6%)	
**Income** **Quartile**	**Quartile 1**	5514 (32.1%)	5924 (31.5%)	0.580	5852 (30.3%)	7595 (30.9%)	0.320
	**Quartile 2**	4801 (27.9%)	5242 (27.9%)		5527 (28.6%)	6882 (28.0%)	
	**Quartile 3**	3953 (23.0%)	4392 (23.3%)		4497 (23.3%)	5771 (23.5%)	
	**Quartile 4**	2927 (17.0%)	3262 (17.3%)		3462 (17.9%)	4328 (17.6%)	
**Insurance**	**Private Insurance**	7002 (39.7%)	8254 (43.1%)	**<0.001**	1394 (7.1%)	1310 (5.2%)	**<0.001**
	**Medicare**	5233 (29.6%)	4860 (25.4%)		17,819 (90.4%)	23,141 (92.6%)	
	**Medicaid**	3136 (17.8%)	4147 (21.6%)		196 (1.0%)	330 (1.3%)	
	**Self-Pay**	1412 (8.0%)	1208 (6.3%)		67 (0.3%)	62 (0.2%)	
	**No Charge**	137 (0.8%)	106 (0.6%)		7 (0.0%)	9 (0.0%)	
	**Other**	730 (4.1%)	589 (3.1%)		223 (1.1%)	138 (0.6%)	
**Hospital** **Location**	**Rural**	3015 (17.0%)	3381 (17.6%)	0.360	4270 (21.6%)	5368 (21.5%)	**0.015**
	**Urban: Non-Teaching**	7862 (44.4%)	8480 (44.1%)		9026 (45.7%)	11,762 (47.0%)	
	**Urban: Teaching**	6825 (38.6%)	7353 (38.3%)		6444 (32.6%)	7888 (31.5%)	
**Comorbidities**	**AIDS**	159 (0.9%)	51 (0.3%)	**<0.001**	14 (0.1%)	2 (0.0%)	**0.001**
	**Alcohol Abuse**	1050 (5.9%)	384 (2.0%)	**<0.001**	402 (2.0%)	137 (0.5%)	**<0.001**
	**Deficiency Anemias**	2215 (12.5%)	3356 (17.5%)	**<0.001**	3967 (20.1%)	5392 (21.6%)	**<0.001**
	**Rheumatoid Arthritis**	200 (1.1%)	821 (4.3%)	**<0.001**	336 (1.7%)	1128 (4.5%)	**<0.001**
	**Chronic Blood Loss**	74 (0.4%)	120 (0.6%)	**0.006**	164 (0.8%)	216 (0.9%)	0.710
	**Congestive Heart Failure**	708 (4.0%)	656 (3.4%)	**0.003**	3190 (16.2%)	4169 (16.7%)	0.150
	**Chronic Pulmonary Disease**	2439 (13.8%)	3456 (18.0%)	**<0.001**	4992 (25.3%)	5929 (23.7%)	**<0.001**
	**Coagulopathy**	596 (3.4%)	479 (2.5%)	**<0.001**	821 (4.2%)	631 (2.5%)	**<0.001**
	**Depression**	1799 (10.2%)	3442 (17.9%)	**<0.001**	1605 (8.1%)	3200 (12.8%)	**<0.001**
	**Diabetes, Uncomplicated**	2768 (15.6%)	2644 (13.8%)	**<0.001**	4593 (23.3%)	5173 (20.7%)	**<0.001**
	**Diabetes, Chronic Complications**	615 (3.5%)	629 (3.3%)	0.290	838 (4.2%)	905 (3.6%)	**0.001**
	**Drug Abuse**	1096 (6.2%)	1133 (5.9%)	0.240	155 (0.8%)	254 (1.0%)	**0.011**
	**Hypertension**	7155 (40.4%)	6514 (33.9%)	**<0.001**	12,524 (63.4%)	16,723 (66.8%)	**<0.001**
	**Hypothyroidism**	918 (5.2%)	2380 (12.4%)	**<0.001**	2038 (10.3%)	5497 (22.0%)	**<0.001**
	**Liver Disease**	1194 (6.7%)	854 (4.4%)	**<0.001**	464 (2.4%)	414 (1.7%)	**<0.001**
	**Lymphoma**	157 (0.9%)	129 (0.7%)	**0.018**	349 (1.8%)	308 (1.2%)	**<0.001**
	**Fluid/Electrolyte Disorders**	6519 (36.8%)	7604 (39.6%)	**<0.001**	9000 (45.6%)	12,787 (51.1%)	**<0.001**
	**Metastatic Cancer**	870 (4.9%)	1130 (5.9%)	**<0.001**	1176 (6.0%)	1177 (4.7%)	**<0.001**
	**Other Neurological Disorders**	2177 (12.3%)	2011 (10.5%)	**<0.001**	2610 (13.2%)	2918 (11.7%)	**<0.001**
	**Obesity**	1344 (7.6%)	1838 (9.6%)	**<0.001**	1003 (5.1%)	1471 (5.9%)	**<0.001**
	**Paralysis**	1876 (10.6%)	988 (5.1%)	**<0.001**	1036 (5.2%)	739 (3.0%)	**<0.001**
	**Peripheral Vascular Disorders**	458 (2.6%)	347 (1.8%)	**<0.001**	2052 (10.4%)	1768 (7.1%)	**<0.001**
	**Psychoses**	1333 (7.5%)	1615 (8.4%)	**0.002**	619 (3.1%)	974 (3.9%)	**<0.001**
	**Pulmonary Circulation Disorders**	132 (0.7%)	187 (1.0%)	**0.018**	361 (1.8%)	544 (2.2%)	**0.010**
	**Renal Failure**	1259 (7.1%)	969 (5.0%)	**<0.001**	3288 (16.7%)	2762 (11.0%)	**<0.001**
	**Solid Tumor**	513 (2.9%)	639 (3.3%)	**0.018**	1050 (5.3%)	932 (3.7%)	**<0.001**
	**Peptic Ulcer**	11 (0.1%)	14 (0.1%)	0.690	11 (0.1%)	21 (0.1%)	0.270
	**Valvular Disease**	243 (1.4%)	363 (1.9%)	**<0.001**	1036 (5.2%)	1415 (5.7%)	0.060
	**Weight Loss**	1044 (5.9%)	1193 (6.2%)	0.210	1629 (8.3%)	2242 (9.0%)	**0.008**
**Invasive Diagnostic Procedure**		1689 (9.5%)	2017 (10.5%)	**0.002**	2238 (11.3%)	2390 (9.6%)	**<0.001**
**Surgical Procedure**		1079 (6.1%)	922 (4.8%)	**<0.001**	1341 (6.8%)	1205 (4.8%)	**<0.001**
**Invasive or Surgical Procedure**		2476 (14.0%)	2690 (14.0%)	0.970	3122 (15.8%)	3182 (12.7%)	**<0.001**
**Deceased**		146 (0.8%)	104 (0.5%)	**0.001**	615 (3.1%)	741 (3.0%)	**0.350**
		**Mean (SD)**	**Mean (SD)**	** *p* **	**Mean (SD)**	**Mean (SD)**	** *p* **
**Age, Years**		48.63 (11.65)	48.04 (11.40)	**<0.001**	77.65 (7.81)	79.67 (8.22)	**<0.001**
**Modified Frailty Index**		1.08 (1.02)	0.97 (1.01)	**<0.001**	1.71 (1.10)	1.61 (1.06)	**<0.001**
**Time to Invasive Diagnostic Procedure, Days**		3.40 (3.55)	3.41 (3.23)	0.970	3.61 (3.68)	3.96 (4.11)	**0.005**
**Time to Surgical Procedure, Days**		3.95 (6.54)	4.00 (4.98)	0.860	4.45 (4.73)	5.06 (5.53)	**0.005**
**Hospital Length of Stay, Days**		4.41 (5.87)	4.35 (4.57)	0.300	5.29 (5.28)	5.24 (4.84)	0.290
**Total Charges, Dollars**		24,219 (41,680)	23,302 (35,865)	**0.025**	27,458 (38,898)	25,849 (35,863)	**<0.001**

Bold indicates result is statistically significant.

### 3.3. Mortality

Only 0.7% of adult patients died in the hospital, compared to the 3.0% of elderly patients who died in the hospital. The deceased adult patients were more than five years older than the surviving adult patients, and the deceased elderly patients were nearly more than three years older than the surviving elderly patients (Table 3). Multiple comorbidity differences were noted when comparing the deceased to the surviving adult or elderly patients. Among the deceased adults, there were higher rates of fluid/electrolyte disorders, congestive heart failure, pulmonary circulation disorders, paralysis, coagulopathy, neurological disorders, liver disease, renal failure, peripheral vascular disorders, valvular diseases, metastatic cancer, weight loss and deficiency anemia (Table 3). Deceased elderly patients exhibited higher rates of congestive heart failure, chronic pulmonary disease, coagulopathy, fluid/electrolyte disorders, metastatic cancer, other neurological disorders, peripheral vascular disorders, renal failure and weight loss (Table 3). The deceased patients from both adult and elderly patients also demonstrated higher rates of invasive diagnostic procedures, surgical procedures, invasive or surgical procedures, time to invasive diagnostic procedures, time to surgical procedures and longer hospital length of stay (HLOS) (Table 3).

**Table 3 ijerph-19-09905-t003:** Characteristics of emergency-admitted patients with the primary diagnosis of paralytic ileus (NIS 2005–2014). Data are classified according to outcome categories.

		Adult, N (%)			Elderly, N (%)		
		Survived	Deceased	*p*	Survived	Deceased	*p*
**All Cases**		36,661 (99.3%)	250 (0.7%)		43,388 (97.0%)	1356 (3.0%)	
**Sex, Female**		19,097 (52.1%)	104 (41.6%)	**0.001**	24,265 (55.9%)	741 (54.6%)	0.350
**Race**	**White**	21,683 (70.4%)	136 (67.0%)	0.110	28,864 (79.7%)	922 (82.2%)	**0.006**
	**Black**	5160 (16.7%)	39 (19.2%)		3691 (10.2%)	120 (10.7%)	
	**Hispanic**	2636 (8.6%)	13 (6.4%)		2212 (6.1%)	40 (3.6%)	
	**Asian/Pacific Islander**	483 (1.6%)	5 (2.5%)		609 (1.7%)	14 (1.2%)	
	**Native American**	221 (0.7%)	1 (0.5%)		215 (0.6%)	3 (0.3%)	
	**Other**	626 (2.0%)	9 (4.4%)		617 (1.7%)	22 (2.0%)	
**Income** **Quartile**	**Quartile 1**	11,350 (31.7%)	85 (35.1%)	0.500	13,022 (30.6%)	424 (32.1%)	0.600
	**Quartile 2**	9978 (27.9%)	63 (26.0%)		12,043 (28.3%)	359 (27.2%)	
	**Quartile 3**	8296 (23.2%)	49 (20.2%)		9963 (23.4%)	299 (22.7%)	
	**Quartile 4**	6143 (17.2%)	45 (18.6%)		7552 (17.7%)	238 (18.0%)	
**Insurance**	**Private Insurance**	15,182 (41.5%)	78 (31.2%)	**0.002**	2628 (6.1%)	77 (5.7%)	0.180
	**Medicare**	9990 (27.3%)	95 (38.0%)		39,715 (91.7%)	1231 (91.1%)	
	**Medicaid**	7228 (19.8%)	54 (21.6%)		506 (1.2%)	19 (1.4%)	
	**Self-Pay**	2607 (7.1%)	12 (4.8%)		122 (0.3%)	7 (0.5%)	
	**No Charge**	241 (0.7%)	2 (0.8%)		15 (0.0%)	1 (0.1%)	
	**Other**	1311 (3.6%)	9 (3.6%)		344 (0.8%)	17 (1.3%)	
**Hospital** **Location**	**Rural**	6358 (17.3%)	35 (14.0%)	0.054	9366 (21.6%)	269 (19.8%)	0.054
	**Urban: Non-Teaching**	16,238 (44.3%)	101 (40.4%)		20,168 (46.5%)	614 (45.3%)	
	**Urban: Teaching**	14,065 (38.4%)	114 (45.6%)		13,854 (31.9%)	473 (34.9%)	
**Comorbidities**	**AIDS**	206 (0.6%)	4 (1.6%)	0.060	16 (0.0%)	0 (0%)	0.999
	**Alcohol Abuse**	1419 (3.9%)	14 (5.6%)	0.160	520 (1.2%)	19 (1.4%)	0.500
	**Deficiency Anemias**	5502 (15.0%)	66 (26.4%)	**<0.001**	9078 (20.9%)	278 (20.5%)	0.710
	**Rheumatoid Arthritis**	1013 (2.8%)	6 (2.4%)	0.730	1427 (3.3%)	36 (2.7%)	0.200
	**Chronic Blood Loss**	188 (0.5%)	5 (2.0%)	**0.001**	360 (0.8%)	19 (1.4%)	**0.024**
	**Congestive Heart Failure**	1319 (3.6%)	45 (18.0%)	**<0.001**	6908 (15.9%)	445 (32.8%)	**<0.001**
	**Chronic Pulmonary Disease**	5850 (16.0%)	43 (17.2%)	0.590	10,503 (24.2%)	413 (30.5%)	**<0.001**
	**Coagulopathy**	1040 (2.8%)	35 (14.0%)	**<0.001**	1352 (3.1%)	100 (7.4%)	**<0.001**
	**Depression**	5205 (14.2%)	30 (12.0%)	0.320	4707 (10.8%)	97 (7.2%)	**<0.001**
	**Diabetes, Uncomplicated**	5372 (14.7%)	38 (15.2%)	0.810	9516 (21.9%)	245 (18.1%)	**0.001**
	**Diabetes, Chronic Complications**	1234 (3.4%)	9 (3.6%)	0.840	1682 (3.9%)	61 (4.5%)	0.240
	**Drug Abuse**	2216 (6.0%)	11 (4.4%)	0.280	400 (0.9%)	8 (0.6%)	0.210
	**Hypertension**	13,567 (37.0%)	99 (39.6%)	0.400	28,468 (65.6%)	769 (56.7%)	**<0.001**
	**Hypothyroidism**	3275 (8.9%)	22 (8.8%)	0.940	7340 (16.9%)	192 (14.2%)	**0.008**
	**Liver Disease**	2010 (5.5%)	36 (14.4%)	**<0.001**	839 (1.9%)	39 (2.9%)	**0.014**
	**Lymphoma**	284 (0.8%)	2 (0.8%)	0.720	626 (1.4%)	31 (2.3%)	**0.011**
	**Fluid/Electrolyte Disorders**	13,955 (38.1%)	157 (62.8%)	**<0.001**	20,931 (48.2%)	845 (62.3%)	**<0.001**
	**Metastatic Cancer**	1918 (5.2%)	80 (32.0%)	**<0.001**	2213 (5.1%)	140 (10.3%)	**<0.001**
	**Other Neurological Disorders**	4143 (11.3%)	37 (14.8%)	0.080	5315 (12.2%)	209 (15.4%)	**<0.001**
	**Obesity**	3165 (8.6%)	12 (4.8%)	**0.031**	2420 (5.6%)	51 (3.8%)	**0.004**
	**Paralysis**	2831 (7.7%)	31 (12.4%)	**0.006**	1715 (4.0%)	58 (4.3%)	0.550
	**Peripheral Vascular Disorders**	791 (2.2%)	14 (5.6%)	**<0.001**	3654 (8.4%)	164 (12.1%)	**<0.001**
	**Psychoses**	2931 (8.0%)	15 (6.0%)	0.250	1555 (3.6%)	39 (2.9%)	0.170
	**Pulmonary Circulation Disorders**	305 (0.8%)	14 (5.6%)	**<0.001**	847 (2.0%)	57 (4.2%)	**<0.001**
	**Renal Failure**	2180 (5.9%)	46 (18.4%)	**<0.001**	5727 (13.2%)	319 (23.5%)	**<0.001**
	**Solid Tumor**	1143 (3.1%)	9 (3.6%)	0.660	1908 (4.4%)	73 (5.4%)	0.080
	**Peptic Ulcer**	25 (0.1%)	0 (0%)	0.999	30 (0.1%)	2 (0.1%)	0.250
	**Valvular Disease**	599 (1.6%)	7 (2.8%)	0.150	2358 (5.4%)	91 (6.7%)	**0.042**
	**Weight Loss**	2165 (5.9%)	71 (28.4%)	**<0.001**	3593 (8.3%)	275 (20.3%)	**<0.001**
**Invasive Diagnostic Procedure**		3667 (10.0%)	37 (14.8%)	**0.012**	4469 (10.3%)	158 (11.7%)	0.110
**Surgical Procedure**		1929 (5.3%)	70 (28.0%)	**<0.001**	2355 (5.4%)	189 (13.9%)	**<0.001**
**Invasive or Surgical Procedure**		5074 (13.8%)	88 (35.2%)	**<0.001**	6011 (13.9%)	290 (21.4%)	**<0.001**
		**Mean (SD)**	**Mean (SD)**	** *p* **	**Mean (SD)**	**Mean (SD)**	** *p* **
**Age, Years**		48.28 (11.53)	53.98 (9.03)	**<0.001**	78.70 (8.09)	81.54 (8.10)	**<0.001**
**Modified Frailty Index**		1.02 (1.01)	1.70 (1.03)	**<0.001**	1.64 (1.07)	1.96 (1.17)	**<0.001**
**Time to Invasive Diagnostic Procedure, Days**		3.36 (3.28)	8.80 (7.55)	**<0.001**	3.68 (3.63)	6.82 (8.25)	**<0.001**
**Time to Surgical Procedure, Days**		3.88 (5.73)	6.83 (8.43)	**0.010**	4.66 (5.05)	5.73 (5.96)	**0.023**
**Hospital Length of Stay, Days**		4.32 (5.00)	12.48 (17.91)	**<0.001**	5.17 (4.80)	8.08 (9.71)	**<0.001**
**Total Charges, Dollars**		23,230 (36,190)	99,500 (157,102)	**<0.001**	25,632 (33,260)	56,101 (97,159)	**<0.001**

Bold indicates result is statistically significant.

### 3.4. Operation vs. No Operation

The stratified analysis based on operation is presented in Table 4. Among the total number of adult patients, 2001 (5.4%) had an operation, and among the total number of elderly patients, 2546 (5.7%) had an operation. The mean age of the adult group who had operations was 2.27 years older than the group with no operations (Table 4), and the mean age of the elderly group with operations was 1.01 years younger than the group with no operation (Table 4). In both non-operatively managed adult and elderly groups, most patients were females, White, funded by private insurance and admitted to urban non-teaching hospitals. In adult and elderly groups with operations, the rates of most comorbidities were higher than in the groups without operations. Furthermore, patients with operations had higher rates of invasive diagnostic procedures, death and longer HLOS as compared to the non-surgical group (Table 4).

**Table 4 ijerph-19-09905-t004:** Characteristics of emergency-admitted patients with the primary diagnosis of paralytic ileus (NIS 2005–2014). Data are stratified according to operation status.

		Adult, N (%)			Elderly, N (%)		
		No Operation	Operation	*p*	No Operation	Operation	*p*
**All Cases**		34,939 (94.6%)	2001 (5.4%)		42,217 (94.3%)	2546 (5.7%)	
**Sex, Female**		18,292 (52.4%)	922 (46.1%)	**<0.001**	23,813 (56.4%)	1205 (47.3%)	**<0.001**
**Race**	**White**	20,629 (70.3%)	1213 (71.4%)	0.380	28,067 (79.9%)	1732 (78.4%)	**0.007**
	**Black**	4911 (16.7%)	290 (17.1%)		3538 (10.1%)	273 (12.4%)	
	**Hispanic**	2520 (8.6%)	130 (7.6%)		2122 (6.0%)	132 (6.0%)	
	**Asian/Pacific Islander**	462 (1.6%)	26 (1.5%)		589 (1.7%)	34 (1.5%)	
	**Native American**	216 (0.7%)	6 (0.4%)		212 (0.6%)	6 (0.3%)	
	**Other**	600 (2.0%)	35 (2.1%)		606 (1.7%)	33 (1.5%)	
**Income** **Quartile**	**Quartile 1**	10,859 (31.8%)	583 (30.1%)	0.440	12,719 (30.7%)	729 (29.2%)	0.170
	**Quartile 2**	9491 (27.8%)	555 (28.7%)		11,709 (28.3%)	701 (28.1%)	
	**Quartile 3**	7896 (23.2%)	454 (23.4%)		9642 (23.3%)	626 (25.1%)	
	**Quartile 4**	5853 (17.2%)	345 (17.8%)		7350 (17.7%)	443 (17.7%)	
**Insurance**	**Private Insurance**	14,488 (41.6%)	784 (39.3%)	**0.045**	2543 (6.0%)	162 (6.4%)	0.890
	**Medicare**	9496 (27.3%)	599 (30.0%)		38,638 (91.7%)	2326 (91.4%)	
	**Medicaid**	6875 (19.7%)	410 (20.6%)		495 (1.2%)	31 (1.2%)	
	**Self-Pay**	2497 (7.2%)	124 (6.2%)		123 (0.3%)	6 (0.2%)	
	**No Charge**	230 (0.7%)	13 (0.7%)		16 (0.0%)	0 (0%)	
	**Other**	1257 (3.6%)	65 (3.3%)		340 (0.8%)	21 (0.8%)	
**Hospital** **Location**	**Rural**	6171 (17.7%)	225 (11.2%)	**<0.001**	9268 (22.0%)	370 (14.5%)	**<0.001**
	**Urban: Non-Teaching**	15,529 (44.4%)	829 (41.4%)		19,856 (46.4%)	1206 (47.4%)	
	**Urban: Teaching**	13,239 (37.9%)	947 (47.3%)		13,363 (31.7%)	970 (38.1%)	
**Comorbidities**	**AIDS**	195 (0.6%)	15 (0.7%)	0.270	13 (0.0%)	3 (0.1%)	0.060
	**Alcohol Abuse**	1305 (3.7%)	129 (6.4%)	**<0.001**	490 (1.2%)	49 (1.9%)	**<0.001**
	**Deficiency Anemias**	5119 (14.7%)	452 (22.6%)	**<0.001**	8667 (20.5%)	692 (27.2%)	**<0.001**
	**Rheumatoid Arthritis**	958 (2.7%)	63 (3.1%)	0.280	1399 (3.3%)	65 (2.6%)	**0.036**
	**Chronic Blood Loss**	170 (0.5%)	24 (1.2%)	**<0.001**	326 (0.8%)	54 (2.1%)	**<0.001**
	**Congestive Heart Failure**	1256 (3.6%)	108 (5.4%)	**<0.001**	6872 (16.3%)	487 (19.1%)	**<0.001**
	**Chronic Pulmonary Disease**	5572 (15.9%)	323 (16.1%)	0.820	10,295 (24.4%)	626 (24.6%)	0.820
	**Coagulopathy**	932 (2.7%)	143 (7.1%)	**<0.001**	1304 (3.1%)	148 (5.8%)	**<0.001**
	**Depression**	4974 (14.2%)	267 (13.3%)	0.270	4527 (10.7%)	278 (10.9%)	0.760
	**Diabetes, Uncomplicated**	5089 (14.6%)	323 (16.1%)	0.052	9190 (21.8%)	576 (22.6%)	0.310
	**Diabetes, Chronic Complications**	1173 (3.4%)	71 (3.5%)	0.650	1611 (3.8%)	132 (5.2%)	**<0.001**
	**Drug Abuse**	2124 (6.1%)	105 (5.2%)	0.130	387 (0.9%)	22 (0.9%)	0.790
	**Hypertension**	12,881 (36.9%)	790 (39.5%)	**0.019**	27,646 (65.5%)	1602 (62.9%)	**0.008**
	**Hypothyroidism**	3137 (9.0%)	162 (8.1%)	0.180	7152 (16.9%)	383 (15.0%)	**0.013**
	**Liver Disease**	1774 (5.1%)	274 (13.7%)	**<0.001**	730 (1.7%)	148 (5.8%)	**<0.001**
	**Lymphoma**	268 (0.8%)	18 (0.9%)	0.510	634 (1.5%)	23 (0.9%)	**0.015**
	**Fluid/Electrolyte Disorders**	13,211 (37.8%)	913 (45.6%)	**<0.001**	20,313 (48.1%)	1474 (57.9%)	**<0.001**
	**Metastatic Cancer**	1790 (5.1%)	210 (10.5%)	**<0.001**	2131 (5.0%)	222 (8.7%)	**<0.001**
	**Other Neurological Disorders**	3927 (11.2%)	261 (13.0%)	**0.013**	5203 (12.3%)	325 (12.8%)	0.510
	**Obesity**	3010 (8.6%)	172 (8.6%)	0.980	2289 (5.4%)	185 (7.3%)	**<0.001**
	**Paralysis**	2678 (7.7%)	187 (9.3%)	**0.006**	1635 (3.9%)	140 (5.5%)	**<0.001**
	**Peripheral Vascular Disorders**	750 (2.1%)	55 (2.7%)	0.070	3575 (8.5%)	245 (9.6%)	**0.043**
	**Psychoses**	2797 (8.0%)	152 (7.6%)	0.510	1511 (3.6%)	83 (3.3%)	0.400
	**Pulmonary Circulation Disorders**	289 (0.8%)	30 (1.5%)	**0.002**	832 (2.0%)	73 (2.9%)	**0.002**
	**Renal Failure**	2018 (5.8%)	210 (10.5%)	**<0.001**	5610 (13.3%)	440 (17.3%)	**<0.001**
	**Solid Tumor**	1061 (3.0%)	91 (4.5%)	**<0.001**	1841 (4.4%)	141 (5.5%)	**0.005**
	**Peptic Ulcer**	24 (0.1%)	1 (0.0%)	0.999	28 (0.1%)	4 (0.2%)	0.110
	**Valvular Disease**	564 (1.6%)	42 (2.1%)	0.100	2282 (5.4%)	169 (6.6%)	**0.008**
	**Weight Loss**	1875 (5.4%)	362 (18.1%)	**<0.001**	3306 (7.8%)	565 (22.2%)	**<0.001**
**Invasive Diagnostic Procedure**		3166 (9.1%)	541 (27.0%)	**<0.001**	3758 (8.9%)	870 (34.2%)	**<0.001**
**Deceased**		180 (0.5%)	70 (3.5%)	**<0.001**	1167 (2.8%)	189 (7.4%)	**<0.001**
		**Mean (SD)**	**Mean (SD)**	** *p* **	**Mean (SD)**	**Mean (SD)**	** *p* **
**Age, Years**		48.19 (11.56)	50.56 (10.68)	**<0.001**	78.84 (8.13)	77.83 (7.63)	**<0.001**
**Modified Frailty Index**		1.01 (1.01)	1.27 (1.04)	**<0.001**	1.64 (1.08)	1.85 (1.12)	**<0.001**
**Time to Invasive Diagnostic Procedure, Days**		3.22 (3.13)	4.56 (4.50)	**<0.001**	3.52 (3.43)	4.96 (5.43)	**<0.001**
**Hospital Length of Stay, Days**		4.01 (3.96)	10.71 (13.78)	**<0.001**	4.88 (4.24)	11.57 (10.31)	**<0.001**
**Total Charges, Dollars**		21,111 (26,716)	69,609 (114,254)	**<0.001**	24,029 (30,451)	68,622 (84,591)	**<0.001**

Bold indicates result is statistically significant.

### 3.5. Risk Factors of Mortality

The multivariable logistic regression model for mortality was built for the group with operations (Table 5) and compared to the model built for the group with no operations (Table 6). The most significant variables in the group with operations were time to operation, age and modified frailty index. For operated adult and elderly patients, each additional day to operation increased the odds of mortality by 4.1% and 3.2%, respectively. Regarding age, every additional year would increase the odds of mortality by 3.8% for operated adult patients, while increasing by 2.6% for operated elderly patients. The most significant variables in the group with no operations were hospital length of stay, age, modified frailty index, sex and invasive diagnostic procedure. However, sex was not a risk factor for non-operated elderly patients, and the invasive diagnostic procedure was not a risk factor for non-operated adults (Table 6). For non-operated patients, for each additional day of hospitalization, the odds of mortality increased by 7.6%, and 5.8% among adult and elderly patients, respectively. Regarding age, every additional year would increase the odds of mortality by 4.0% for non-operated adult patients, while increasing by 5.0% for non-operated elderly patients. Being female decreased the odds of mortality by 28.4% among non-operated adult patients; however, sex did not play a significant role among non-operated elderly patients. As a result of invasive diagnostic procedures, the odds of mortality decreased by 21.6% among non-operated elderly patients while not being significant among non-operated adult patients (Table 6).

**Table 5 ijerph-19-09905-t005:** Backward multivariable logistic regression analysis to evaluate the associations between mortality and different factors in emergency-admitted patients with the primary diagnosis of paralytic ileus and undergoing an operation (NIS 2005–2014). Mortality was the dependent variable.

	Adult Operation		Elderly Operation	
	*n* = 1730	R^2^ = 0.073	*n* = 2230	R^2^ = 0.013
	OR (95% CI)	*p*	OR (95% CI)	*p*
**Time to Operation, Days**	1.041 (1.015, 1.068)	**0.002**	1.032 (1.006, 1.059)	**0.014**
**Age, Years**	1.038 (1.005, 1.073)	**0.026**	1.026 (1.005, 1.047)	**0.015**
**Modified Frailty Index**	1.529 (1.207, 1.936)	**<0.001**	**Removed Via** **Backward** **Elimination**	
**Sex, Female**	**Removed Via** **Backward** **Elimination**			
**Invasive Diagnostic Procedure**				
**Race**				
**Income Quartile**				
**Insurance**				
**Hospital Location**				

Bold indicates result is statistically significant.

**Table 6 ijerph-19-09905-t006:** Backward multivariable logistic regression analysis to evaluate the associations between mortality and different factors in emergency-admitted patients with the primary diagnosis of paralytic ileus and not undergoing an operation (NIS 2005–2014). Mortality was the dependent variable.

	Adult Non-Operation		Elderly Non-Operation	
	*n* = 34,888	R^2^ = 0.068	*n* = 42,200	R^2^ = 0.041
	OR (95% CI)	*p*	OR (95% CI)	*p*
**Hospital Length of Stay, Days**	1.076 (1.059, 1.094)	**<0.001**	1.058 (1.049, 1.068)	**<0.001**
**Age, Years**	1.040 (1.023, 1.058)	**<0.001**	1.050 (1.042, 1.057)	**<0.001**
**Modified Frailty Index**	1.392 (1.220, 1.590)	**<0.001**	1.261 (1.197, 1.330)	**<0.001**
**Sex, Female**	0.716 (0.531, 0.965)	**0.028**	**Removed**	
**Invasive Diagnostic Procedure**	**Removed Via** **Backward** **Elimination**		0.784 (0.633, 0.971)	**0.026**
**Race**			**Removed Via** **Backward** **Elimination**	
**Income Quartile**				
**Insurance**				
**Hospital Location**				

Bold indicates result is statistically significant.

The modified frailty index was estimated based upon the variables available within the NIS dataset as described in the methods. Elderly patients managed operatively showed no significant association between mortality and frailty (Table 6). Operatively treated frail adults demonstrated a 52.9% increased risk of mortality relative to non-frail adults. In the non-operative groups, elderly and adult frail patients exhibited 26.1% and 39.2% increased odds of mortality, respectively (Table 5). A comparison of variables utilized to estimate the five-item modified frailty index is shown in Table 7. Lastly, Figure 1 is a graphic representation of the major mortality risk factors (Hospital Length of Stay, Time to Surgery, Frailty, and Age) and protective factors (Female Sex and Invasive Diagnostic Procedure) in patients emergently admitted with paralytic ileus.

**Table 7 ijerph-19-09905-t007:** Comparison of variables used for estimating 5-item modified frailty index (mFI).

Subramaniam 5-Item MFI Factors	NIS Variable for Our Modified 5-Item mFI Estimate
Functional Health Status (Partially or Totally)	Presence of at Least 1: Solid Tumor, Renal Failure, Metastatic Cancer, Paralysis, Lymphoma, Coagulopathy, Weight Loss
Diabetes Mellitus (Non/Insulin)	Diabetes (Uncomplicated) or Diabetes (Chronic Complications)
History of COPD	Chronic Pulmonary Disease
History of Congestive Heart Failure	Congestive Heart Failure
Hypertension Requiring Medication	Hypertension (Un/Complicated)

## 4. Discussion

### 4.1. Age and Mortality

Our results reveal mortality rates of 3.0% and 0.7% in elderly and adult patients with paralytic ileus, respectively. This finding illustrates that elderly patients displayed a mortality risk more than four-fold higher than their adult patient counterparts. Elevated mortality rates were observed in elderly patients in comparison to adult patients in both the surgical and non-surgical groups. The current literature provides a mixture of supporting and negating results regarding the relationship between age, mortality and ileus.

After conducting a 10-year retrospective analysis on elderly and adult patients with ileus, Koşar and Görgülü demonstrated higher mortality rates in elderly relative to adult patients [7]. In a similar analysis, Kapan et al. illustrated advanced age to be associated with higher rates of mortality in patients with bowel obstructions [11]. Contributing further to these findings, Narayana and Sharma completed a comparative study that revealed significantly higher mortality rates in older and middle-aged people compared to younger people with small or large bowel obstructions [12]. Contrary to these results, Ogunrinde et al. assessed 217 patients with intestinal obstructions and found no statistically significant association between age and clinical outcome [8]. In addition, Arenal et al. analyzed mortality-related factors in elderly patients who underwent operations for intestinal obstruction. The analysis conducted by Arenal et al. revealed no significant association between older age, morbidity or mortality in patients undergoing surgery for intestinal obstruction [13].

These findings demonstrate that the association between mortality, ileus and age has yet to be fully elucidated. We recommend further large-scale analyses controlling for a larger proportion of patient attributes to appropriately address this incongruency. Determining the precise association between age, ileus and mortality in patients with ileus can provide an easily accessible prognostic factor for clinical decision making.

### 4.2. Hospital Length of Stay and Mortality

Hospital length of stay (HLOS) has been indicated as a major mortality factor in a number of emergent medical ailments. Our findings show that for every additional day non-operatively managed adult and elderly patients with paralytic ileus stayed in the hospital, the odds of mortality increases by 7.6% and 5.8%, respectively. Gifford et al. conducted a four-year retrospective review to determine the factors associated with postoperative ileus (POI). Their study concluded that liver disease, higher intraoperative morphine utilization, advanced age and history of substance abuse were significant predictors of POI [14]. Moreover, Gifford et al. demonstrated POI was associated with reduced patient satisfaction, increased healthcare cost and increased HLOS [14]. In a similar analysis, Greenberg et al. analyzed the risk factors and outcomes associated with POI. Their study revealed associations between POI, increased patient discomfort, longer HLOS and higher healthcare cost. Greenberg et al. further demonstrated significant associations between POI, higher patient BMI, longer procedure duration and a postoperative day 2 net fluid balance greater than 800 mL [15].

These findings emphasize modifiable factors influencing postoperative ileus, which in turn increases HLOS and the odds of mortality. Intraoperative morphine utilization, fluid management and procedure duration should be further evaluated to assess their ability to reduce POI, HLOS and mortality.

In regard to perioperative fluid administration, Gupta and Gan recommended individualized fluid management plans utilizing a zero-balance fluid approach with salt crystalloid solutions followed by postoperative eating and drinking encouragement [16]. These recommendations are further supported by Chowdhury and Lobo who concluded individualized intraoperative fluid control utilizing a zero-fluid balance approach reduces time to return of normal gastrointestinal function [17]. By reducing the gastrointestinal recovery time, providers can significantly reduce HLOS. Another technique to reduce HLOS that is under current investigation is the utilization of NSAID therapeutics in place of opioids for perioperative analgesia. A randomized controlled trial (RCT) concluded COX-2 specific inhibitors shorten gastrointestinal recovery time following colorectal resection as compared to morphine-only regimens [18]. Similarly, Xu et al. conducted an RCT demonstrating that perioperative flurbiprofen axetil facilitated a reduced recovery time of bowel function, enhanced analgesia and attenuated cytokine response [19]. In addition, an RCT involving 102 patients undergoing colorectal resection exhibited the earlier restoration of normal bowel function with ketorolac and intravenous patient-controlled morphine (IVPCM) as opposed to IVPCM without ketorolac [20]. These studies put forth excellent evidence that should be employed to reduce POI and HLOS in patients undergoing colorectal operations.

Reducing HLOS is of paramount importance to reduce the risk of nosocomial infections. Murni et al. completed a 3.5-year prospective study that revealed HLOS greater than 7 days significantly increased the risk of healthcare-associated infections [21]. This finding contributes to the importance of reducing HLOS in patients with ileus whose HLOS frequently extends beyond 7 days with an average of 6–11 days [14,15,22,23,24]. The significance of reducing HLOS is supported by studies demonstrating an association between HLOS, mortality and numerous pathologies. An analysis of 33,700 emergently admitted patients with ventral hernia concluded that increased HLOS was a main risk factor for mortality in non-operatively managed patients [25]. Lin et al. revealed significant associations between mortality and increased HLOS in elderly and non-elderly patients hospitalized with chronic duodenal ulcers [26]. An investigation of 52,786 patients demonstrated increased HLOS was a risk factor for mortality in non-operated emergency hemorrhoid patients [27]. Lastly, a large-scale retrospective analysis of 48,539 patients who were managed non-operatively following emergent admission for ventral hernia displayed significant associations between mortality and increased HLOS [28]. Clearly, the association between mortality and increased HLOS has been well established through numerous and robust investigations.

We suspect the increased mortality rates observed in patients with ileus following an increased HLOS may be a consequence of nosocomial infections, higher rates of pathological intestinal complications and higher acuity patients requiring longer HLOS. We recommend further investigations into the aforementioned modifiable factors to reduce HLOS and thereby reduce nosocomial infections, secondary intestinal pathology and mortality. Furthermore, HLOS should be utilized to prognosticate patients hospitalized with paralytic ileus to initiate prophylactic measures.

### 4.3. Modified Frailty Index and Mortality

Frailty has been identified as an independent predictor of postoperative complications, HLOS and mortality in surgically and non-surgically managed patient populations [29,30,31]. In a large-scale retrospective analysis on the impact of frailty following colectomy, Hadaya et al. concluded frailty is associated with mortality, increased healthcare cost, higher rate of re-admission and longer HLOS [32]. Frailty has also been shown to be associated with a 59% relative increased risk of postoperative ileus following colonic resection [33]. In a similar analysis, Venkat et al. determined the modified frailty index independently predicts morbidity, mortality, cardiopulmonary complications and prolonged HLOS in patients with clostridium difficile [34]. The modified frailty index has also been shown to predict morbidity and mortality after pancreaticoduodenectomy after adjustment for age, sex, albumin and body mass index [35]. Additionally, a 2-year analysis of colorectal cancer patients illustrated a high modified frailty index is associated with increased odds of delayed gastrointestinal function recovery, longer HLOS and POI [36].

Our results echo these conclusions by revealing that the five-item modified frailty index can be implemented to predict mortality in adult and elderly patients emergently admitted with paralytic ileus. Specifically, in the operatively managed group of adult frail patients, there was a 52.9% increased odds of mortality relative to non-frail adults. Similarly, in the non-operatively managed group of adult and elderly frail patients, there was a 39.2% and 26.1% increased odds of mortality, respectively. These results are specific to our estimated five-item modified frailty index because the NIS does not contain all variables required to calculate the Subramaniam five-factor modified frailty index [31]. The estimations employed to determine the five-item modified frailty index with the variables available within the dataset is described within the aforementioned methods.

Overall, our results and the supporting literature emphasize the utility of the modified frailty index as a prognosticating tool for patients with paralytic ileus. In the future, it is critical to address clinically modifiable factors influencing frailty to improve clinical outcomes in perioperative and non-operatively managed settings. We recommend further investigations into the efficacy of therapeutics aimed toward reducing patient frailty following emergent admission for paralytic ileus.

### 4.4. Time to Operation and Mortality

Our analysis shows that for every one-day delay in performing an operation in adult and elderly patients with paralytic ileus, there is an increase in mortality odds by 4.1% and 3.2%, respectively. Based upon our literature review, no studies have established a relationship between time to operation (TTO) and mortality in patients with paralytic ileus. Although relatively under-investigated in the setting of ileus, a longer TTO has been well established as a mortality and morbidity risk factor. A prospective analysis of patients undergoing emergent general surgery concluded in-hospital surgical delays were associated with higher mortality risk [37]. Leeds et al. investigated the impact of surgical timing in 76,364 patients undergoing emergent hernia surgery and concluded that a delay in surgery increased the odds of morbidity, HLOS and 30-day mortality risk [38]. Similarly, a retrospective study on the predictors of mortality in patients emergently admitted for arterial embolism and thrombosis revealed delayed operating-room access was associated with increased HLOS and mortality [39]. An additional study by Levy et al. further demonstrated delayed TTO was associated with increased mortality odds in patients emergently admitted with empyema [40]. Our results and the supporting literature emphasize the critical association present between TTO and mortality that is also evident in patients emergently admitted with paralytic ileus. Further research is required to identify the pathogenesis by which mortality odds increase following delayed surgical intervention.

While the exact mechanism between TTO and mortality has yet to be established, it is important to note TTO is a highly variable factor influenced by modifiable clinical and logistical features. Chagpar et al. conducted a ten-center analysis to identify factors influencing time to surgery in breast cancer patients. Their study concluded that a preoperative MRI and community practice setting as opposed to academic setting were independent predictors of delayed time to operation [41]. Additional causes of operative delay included equipment failure, surgeon unavailability, patient clinical condition and availability of instruments [42,43,44]. We recommend additional studies aimed at mitigating factors delaying TTO as a potential mortality-reducing intervention in patients emergently admitted with paralytic ileus.

### 4.5. Sex and Mortality

Previous studies have indicated that males develop ileus at significantly higher rates than females [45,46,47,48]. Koch et al. conducted a 3-year retrospective analysis on postoperative ileus that showed significantly higher rates in males following elective colorectal surgery [49]. This finding is supported by Ceretti et al., who displayed significant associations between male gender and ileus [50]. Contrarily, our results reveal higher rates of ileus in adult and elderly females relative to adult and elderly males. No clear evidence has emerged explaining the relationship between ileus and sex. Thus, these findings indicate further investigation is necessary to determine the exact association.

In regard to mortality, our study demonstrated emergently admitted adult female patients with paralytic ileus displayed lower mortality rates than their adult male counterparts. Our finding is in contrast to a 10-year retrospective analysis that demonstrated significantly higher mortality rates in women with intestinal obstructions [7]. Further analysis into the association between sex, ileus and mortality has been under-investigated in the literature, likely due to limited interventional utility. Nevertheless, the enhanced characterization of the relationship may assist clinicians in determining mortality risks in patients with ileus based upon sex differences.

### 4.6. Invasive Diagnostic Procedure and Mortality

Invasive diagnostic procedures may significantly reduce mortality rates in patients emergently admitted with paralytic ileus. Our study revealed invasive diagnostic procedures on the intestines were the most common procedure in both elderly and adult patients (91.8% and 88.7%, respectively). In the elderly group, invasive diagnostic procedures on other aspects of the abdominal region were associated with the highest mortality rate (6.7%). This is in contrast to the adult group whose mortality rate (4.8%) was highest in those who underwent invasive diagnostic procedures on the gallbladder and biliary tract. More importantly, our findings indicate elderly patients who underwent invasive diagnostic procedures exhibit significantly reduced mortality odds.

The clinical utility of invasive diagnostic procedures in patients with paralytic ileus has received growing investigative interest. Our findings support the current research trends exhibiting lower mortality rates associated with invasive diagnostic procedures. This is an important result as previous studies have revealed high mortality rates in patients undergoing emergent surgery for acute ileus [51,52]. Minimally invasive endoscopic techniques may facilitate the decompression of acute ileus and reduce the need for surgery [52]. Transanal tube decompression in the setting of acute colorectal obstruction was shown to have high clinical success rates [53]. Additionally, Vilz et al. concluded that colonoscopic decompression was a viable treatment option in large-bowel ileus and Ogilvie syndrome [54]. Our results and the supporting literature provide evidence toward the utilization of invasive diagnostic procedures in patients with acute ileus. Further enquiries in the form of RCTs are required to determine the relative efficacy of invasive diagnostic procedures in comparison to traditional treatment plans.

### 4.7. Strengths

The results incorporated into our research were obtained from the National Inpatient Sample (NIS). NIS is nationally representative, given the large number of data points that are obtained from multiple hospitals across the United States. In addition, the integration of clinical and societal factors into our analyses allowed for standardization. These factors allowed for statistically strong conclusions that can be generalized to a large population. This study also introduced several avenues for future investigation regarding associations between paralytic ileus invasive diagnostic procedures, TTO, modified frailty index, age, factors influencing HLOS and age. Lastly, our study provided recommendations as to how these results can directly affect clinical outcomes in patients with paralytic ileus.

### 4.8. Limitations

The utilization of the NIS database led to some limitations in this study. The database does not contain all variables required to determine the five-item modified frailty index. As a consequence, we estimated the five-item modified frailty index based upon the results available in the data set. Moreover, the NIS database does not contain information regarding the severity of ileus, etiology, pharmaceutical management, invasive diagnostic procedure protocols, operative techniques utilized and severity of comorbidities. Further stratification of invasive diagnostic procedures, time to operation and disease severity may produce more accurate interpretations of the data.

## 5. Conclusions

Patients emergently admitted for paralytic ileus with increased hospital length of stay, longer time to operation, advanced age or higher modified frailty index displayed higher mortality rates. In non-operatively managed patients with paralytic ileus, adult females and elderly patients who underwent invasive diagnostic procedures both exhibited lower mortality rates. The information drawn from this study could be used to help physicians identify risk factors within their patients that may assist them in proactively managing higher-risk patients. Further research is needed to clarify how to mitigate some of these risk factors and the relationships that may exist between certain risk factors for mortality.

## Figures and Tables

**Figure 1 ijerph-19-09905-f001:**
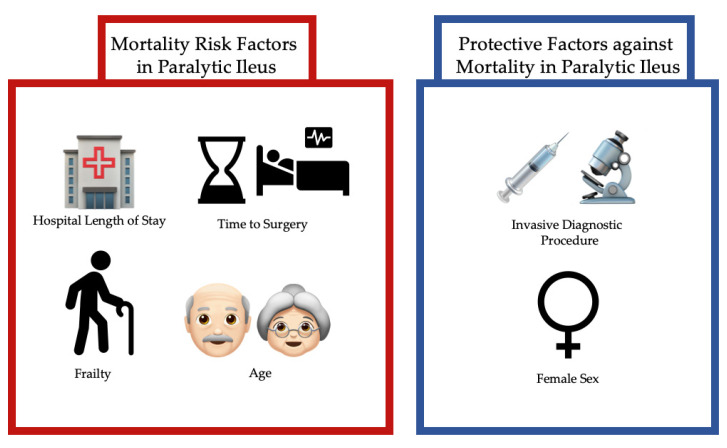
Abstract figure demonstrating main findings of risk factors for and protective factors against mortality in paralytic ileus.

## Data Availability

National inpatient sample database can be found on the Healthcare Cost & Utilization Project website through the following URL, https://www.hcup-us.ahrq.gov/db/nation/nis/nisdbdocumentation.jsp (accessed on 2 July 2022).

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
