# Peer review of "Age Increases the Risk of Mortality by Four-Fold in Patients with Emergent Paralytic Ileus: Hospital Length of Stay, Sex, Frailty, and Time to Operation as Other Risk Factors"

_ijerph, 2022, doi:10.3390/ijerph19169905_

Round 1

Reviewer 1 Report

This paper aims to identify potential predictors of mortality in emergently admitted patients with the primary diagnosis of paralytic ileus through a 10 years retrospective analysis of 81,674 patients. The research topic is interesting, and the main conclusion of this paper shows that patients emergently admitted for paralytic ileus with increased hospital length of stay, longer time to operation, advanced age, or higher modified frailty index display higher mortality rates.

Overall, this paper is well written. However, the research background and motivation of this paper are not very clear. There are several specific issues in the paper need to be addressed. Please see the specific comments as follows.

Specific Comments:

1.      Main academic contributions of this paper are unclear. The authors should state clearly that how the findings or conclusions of this paper could benefit the related research areas.  

2.      For data analysis, there are more than 12 risk factors, how did the authors deal with the categorical variables? Did the authors consider about the potential multi-collinearity problem in the multivariable logistic regression analysis?

3.      A major concern for the risk factor analysis is that the mortality rates were all smaller than 12% (except the Operations on Pancreas case with only 1 deceased in total five patients). Technically, these are imbalanced data regarding mortality rates. Traditional logistic regression can be biased and hence may be improper for the risk factor analysis.

Minor issues:

1.      There are ten keywords in this paper. The number of keywords is usually no more than five. Besides, “age” is not proper for a keyword.

2.      Page 3, row 105, the name of R packages that used for data analysis should be incorporated.

3.      Figures are encouraged to be used for summarizing or visualizing the results.

Reviewer 2 Report

In this study, Elgar and al. are suggesting that Patients emergently admitted for paralytic ileus with increased hospital length of stay, longer time to operation, advanced age, or higher modified frailty index display higher mortality rates. Female sex and invasive diagnostic procedures are negatively correlated with death in non-operatively managed patients with paralytic ileus.

Overall the manuscript is well written and the context is a major of interest. however, there are some minor concerns that need to be improved to be suitable for publication in the current journal.

Minor concerns:

Title:

The title is too long and it’s like a sort of paragraph, therefore it's highly recommended to provide a shorter and more focused one.

Abstract:

the word (Background) is in a bold while, results, .. are not, please unify the form.

Methods:

The methods must be re-arranged and presented in a more organized flow, as in the current form, it’s a description of the provided tables. Therefore, the method section must be subdivided into further sub-sections according to the variety of methods that have been utilized for such evaluation. For instance, a subsection for patient features, statical analysis,.etc.

Results:

Since the study assessed the Mortality rate between 2005-2014, its highly suggested to present a trend or a line graph apart from tables, illustrating the change in mortality rate during 10 years of evaluation (2005-2014)-specifically the Risk Factors of Mortality

Text: P value should be in Latin p, among all the text.

Line 58: please remove (.) after the word (risk).

Introduction: It's highly recommended to mention the global incidence and mortality rate of paralytic ileus at the starting of the introduction. 

Reviewer 3 Report

The manuscript titled: “The risk of mortality in geriatric patients with emergent paralytic ileus is 4-fold greater than that in non-elderly adults: hospital length of stay, sex, frailty, and time to operation as other major risk factors” shows information regarding the impact of several factors with mortality risk in geriatric patients with emergent paralytic ileus. Overall, the manuscript is well written, scientifically sound, hypothesis-driven, and fits within the Journal’s scope. Minor revisions are suggested to make this manuscript suitable for publication in IJERPH.

ABSTRACT

1.     Line 17: Is the National Inpatient Sample Database from 2014?

2.     Lines 21-22: Was the mortality significant? Please add the significance value (e.g., p<0.01).

3.     Line 36: Please delete the word “conclusion”.

KEYWORDS

4.     Please arrange keywords in an alphabetical manner.

INTRODUCTION

5.     Line 47: Why including two references, one of them from 2001 and the other one from 2021?

6.     Line 54: Are these costs for a single patient, admitted to a single hospital entry? Please clarify.

MATERIALS AND METHODS

7.     Line 115: Please spell ICD-9.

8.     Information from Tables 1-5 should be described in the Results section and not the materials section. Please correct it.

9.     Although the authors have indicated it, a statistical analysis section should be indicated.

RESULTS AND DISCUSSION

10.  Line 308: Please correct: “Mortality”.

11.  Some of the recommendations given in the discussion should be reflected in the conclusions.

Reviewer 4 Report

The present manuscript Risk of Mortality in Geriatric Patients with Emergent Paralytic Ileus is 4-fold Greater than that in Non-elderly Adults: Hospital Length of Stay, Sex, Frailty, and Time to Operation as Other Major Risk Factors.  The subject frame of the work is well constructed. So, in this respect and this article should be contributed to present research. I recommended this work for publication after the following minor revisions.

1.      There are several typographical mistakes as well in whole manuscript. Therefore, the author’s thoroughly careful check the language and typo mistake to minimize the error.

2.      The abstract should be beginning with a sentence about the background of concept and the aims as well as novelty of study should be mentions. What exactly is the novelty of this study? The abstract is poorly written and should be improved. Abbreviations must be avoided in abstract. Parenthesis should be avoided in abstract - this is poor writing. Please improve.

3.      Introduction; Check and format the citations in the whole manuscript. Also, Appropriate references must be provided to explained the background, what is already done and why this study carried out. Other vise the novelty of this research is still poorly presented. This is important especially for the high IF journals. The scientific style should be used. What exactly is the aim of this work? Hypothesis statement is missing in the introduction section

4.      All figures are of poor technical quality and not suitable for publication, especially in a high reputed journal. Font size and kind is too small and must be unified in all figures. Small writings are unreadable. All figures must be self-explanatory.

5.      I suggest first time write full name rather than abbreviation; revise throughout in manuscript

Round 2

Reviewer 1 Report

The paper has been improved very well in this revision. The authors have addressed all my

comments. Well done!